# Adaptive Diffusion Priors on Pre-clinical CT Reconstruction

**Theresa Hiu**[*†1,2]                                    THERESA.HIU@TUM.DE
**Daniel Frey**[*1,2] (iD)                                  DANIEL.FREY@TUM.DE
**Tina Dorosti**[1,2,3] (iD)                               TINA.DOROSTI@TUM.DE
**Johannes Thalhammer**[1,2,3,4] (iD)           JOHANNES.THALHAMMER@TUM.DE
**Sebastian Peterhansl**[1,2]                     SEBASTIAN.PETERHANSL@TUM.DE
**Zijin Huang**[1,2]                                        ZJ.HUANG@TUM.DE
**Simon Zandarco**[1,2]                             SIMON.ZANDARCO@TUM.DE
**Franz Pfeiffer**[1,2,3,4] (iD)                        FRANZ.PFEIFFER@TUM.DE
**Florian Schaff**[1,2] (iD)                           FLORIAN.SCHAFF@TUM.DE

[1] *Chair of Biomedical Physics, TUM School of Natural Sciences, TUM*

[2] *Munich Institute of Biomedical Engineering, TUM*

[3] *Institute for Diagnostic and Interventional Radiography, TUM University Hospital, TUM*

[4] *TUM Institute for Advanced Study, TUM*

## Abstract

X-ray dark-field imaging enables visualization of lung microstructure and the detection of pulmonary diseases. However, in pre-clinical micro-CT studies, the reconstruction suffers from severe streak artifacts due to highly undersampled acquisitions constrained by radiation dose. We leverage a diffusion-based reconstruction framework by extending the Deep Diffusion Image Prior (DDIP) to dark-field CT. The method combines a pretrained diffusion prior with physics-based consistency and incorporates low-rank adaptation (LoRA) for improved robustness to out-of-distribution data. Experiments demonstrate increased contrast-to-noise ratio and artifact suppression while preserving edge sharpness, enabling higher-quality dark-field imaging for dose-constrained longitudinal studies.

**Keywords:** Computed tomography, dark-field imaging, diffusion models

## 1. Introduction

X-ray dark-field imaging enables the visualization of the microscopic alveolar structure of lung tissue. Several studies have already demonstrated the capability of dark-field imaging to detect lung diseases such as emphysema, tumors, and radiation-induced lung damage in mice (Yaroshenko et al., 2013; Burkhardt et al., 2021; Gassert et al., 2022). Due to the radiation dose constraints of longitudinal small-animal studies, the data are severely undersampled, with at times as few as 211 cone-beam projections acquired over 360°. Consequently, the reconstructed dark-field images are strongly degraded by streak artifacts.

Diffusion models have demonstrated strong performance in medical image denoising and sparse-view CT reconstruction, but their application to dark-field CT remains largely unexplored. The performance of diffusion models deteriorates under out-of-distribution (OOD) conditions, for example, when scanner characteristics, acquisition protocols, or test data differ from those seen during training.

---

[*] Contributed equally

[†] Corresponding author

In this work, we adapted the Deep Diffusion Image Prior (DDIP) method (Chung and Ye, 2024), originally developed for sparse-view reconstruction under out-of-distribution settings, to small-animal dark-field micro-CT. The approach combines a pretrained diffusion model as an image prior with TIGRE-based (Biguri et al., 2025) forward and backprojected operators for physics-based consistency, alternating direction method of multipliers (ADMM) (Neal et al., 2011) with total variation (TV) regularization applied at each diffusion step to enforce data fidelity, and low-rank adaptation (LoRA) (Hu et al., 2022) for adaptation of the diffusion model to the measured data.

## 2. Methods

Due to the ill-posed nature of the inverse problem, the reconstruction is formulated in a Bayesian framework using a learned image prior. We trained an unconditional diffusion model on $512 \times 512$ synthetic ellipse phantoms (Barbano et al., 2025) with a learning rate of $1 \times 10^{-4}$ for 49 epochs and used it as a learned diffusion prior. To adapt this pretrained model to the measured data, we adopted the DDIP (Chung and Ye, 2024) framework. We further used the DDIP3D (Chung and Ye, 2024) variant incorporating DiffusionMBIR (Chung et al., 2023).

The projections are preprocessed with logarithmic transformation, center-shift correction, and ring artifact reduction, followed by cropping to $300 \times 512$ (rows $\times$ columns) and truncation correction. The TIGRE operator (Biguri et al., 2025) is used to define the forward and adjoint operators. The Feldkamp–Davis–Kress (FDK) reconstruction with a Shepp–Logan filter provides a physics-based initialization, while the adjoint backpropagation is used in data-consistency updates. A reduced denoising diffusion implicit model (DDIM) (Song et al., 2020) sampling schedule with 50 steps is used.

The reconstruction alternates between adaptation and inference for each diffusion step. During adaptation, LoRA parameters are updated online using small randomly sampled slice batches. A projection-domain loss between the forward-projected and measured sinograms is used to update the model, enabling test-time adaptation to the current data. During inference, the model predicts the noise, and a clean estimate is computed via the Tweedie formula (Efron, 2011). This estimate is then refined using an ADMM-based data-consistency step that enforces both measurement fidelity and TV regularization. The resulting estimate is used in the subsequent DDIM update. Iterating this process progressively transforms the physics-informed noisy initialization into the final reconstructed dark-field CT volume.

To further improve the final dark-field image, we performed weighted temporal averaging of intermediate diffusion reconstructions using a shifted sigmoid weighting function. This enables the analysis of contributions across the diffusion trajectory. The shifted parameter $c$ was evaluated from 0 to 20, where increasing $c$ emphasizes later diffusion steps (finer details), while smaller values emphasize earlier stages (global structures), hence $c = 8$ was selected.

## 3. Results and Discussion

We evaluated the proposed method against FDK reconstruction with a Shepp-Logan filter as shown in Fig. 1. The diffusion-based reconstruction achieves improved streak artifact

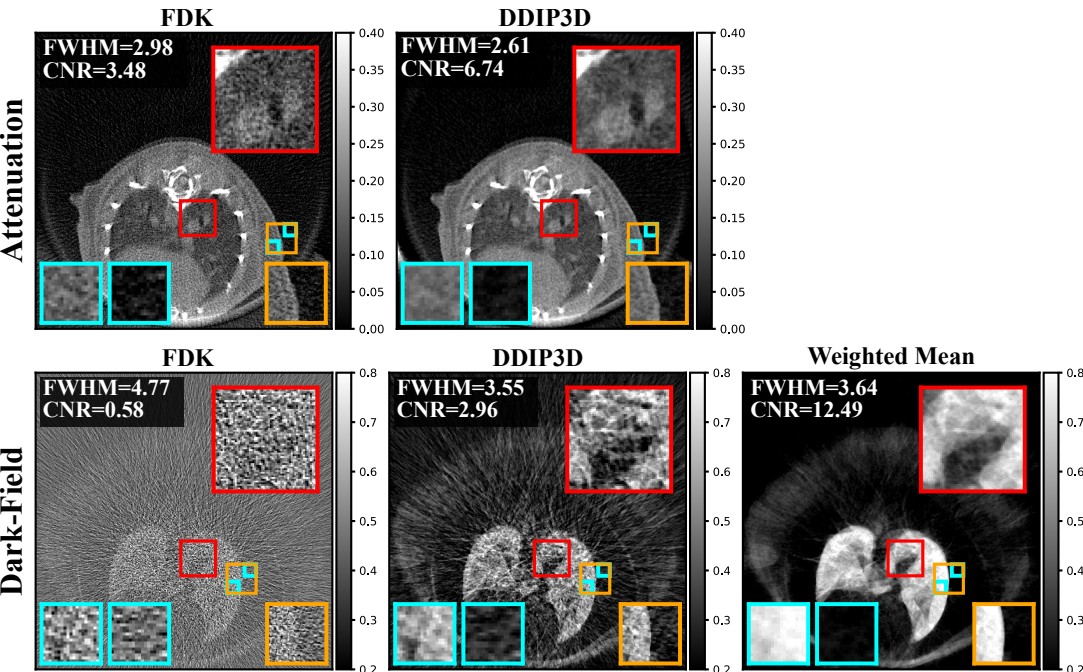

Figure 1: Attenuation and dark-field micro-CT reconstruction of the mouse thoracic cavity. The diffusion-based reconstruction (DDIP3D) recovers fine lung structural details, while the weighted-mean reconstruction suppresses high-frequency artifacts and better preserves the global structure. Lung and air regions (cyan) used for CNR evaluation, and an edge ROI (orange) used for FWHM estimation.

reduction in both attenuation and dark-field images, while preserving edge sharpness and yielding higher CNR. The weighted-mean reconstruction aggregates intermediate diffusion estimates along the reverse trajectory, combining coarse global structure from early diffusion stages with fine details from later stages, which suppresses high-frequency artifacts while preserving anatomical consistency.

This work demonstrates that integrating diffusion models as learned priors with physics-based reconstruction and OOD adaptation provides a powerful framework for addressing the undersampling problem in dark-field CT. The DDIP-based framework achieves improved artifact suppression, enhanced structural fidelity, and higher image quality. These findings demonstrate the potential of diffusion-based methods to advance dark-field imaging in dose-constrained longitudinal pre-clinical studies. Future work will focus on reader studies involving experts to assess the clinical impact of the proposed method.

## Acknowledgments

Financial support through the European Research Council (ERC Smart Detectors for Dark-field X-ray Imaging, SyG 101167328), and the Free State of Bavaria under the Excellence Strategy of the Federal Government and the States, as well as by the Technical University of Munich – Institute for Advanced Study.

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

## Appendix A.

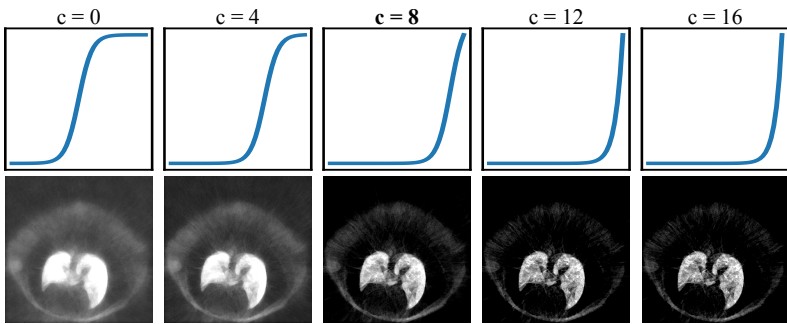

Figure A1: Effect of the sigmoid shift parameter $c$ for weighted temporal averaging. The parameter is varied from 0 to 20, shifting emphasis along the diffusion trajectory. $c = 8$ is selected.

