# OpenReview forum: "Adaptive Diffusion Priors on Pre-clinical CT Reconstruction"
_MIDL.io/2026/Short_Papers — MIDL 2026 - Short Papers Poster_

### Official Review · Reviewer_aP8o · 2026-04-29
**intersting work**

**Rating:** 4
**Confidence:** 4

**Review:**

The paper addresses an important problem in dark-field CT reconstruction under severe undersampling, which is highly relevant for dose-limited pre-clinical imaging. The integration of diffusion priors with physics-based reconstruction and test-time adaptation is technically sound and aligns with recent trends in inverse problems. The methodology is reasonably well-structured, combining DDIP, ADMM-based data consistency, and LoRA adaptation, although the presentation is quite dense and may be challenging for readers unfamiliar with diffusion-based reconstruction frameworks.

In terms of originality, the work is primarily an adaptation and integration of existing components (DDIP, LoRA, ADMM, TIGRE operators) rather than a fundamentally new methodological contribution. However, its application to dark-field CT and the incorporation of weighted temporal averaging provide incremental novelty. T Overall, the work is technically solid, suitable for acceptance as a short paper, and will generate important discussion during the meeting.

**Summary:**

This work proposes a diffusion-based reconstruction framework for dark-field micro-CT that extends the DDIP by integrating physics-based consistency and test-time adaptation via low-rank adaptation (LoRA). Experiments show improved quantitative and qualitative results. Overall, the study demonstrates that diffusion priors combined with physics-based modeling can significantly improve image quality in dose-constrained imaging settings.

**Strengths:**

Addresses a clinically and scientifically important problem: reconstruction under severe undersampling in dark-field CT. Effective integration of diffusion priors with physics-based reconstruction, which is a strong and modern approach to inverse problems. Incorporates test-time adaptation (LoRA), improving robustness to out-of-distribution data—an important practical consideration.

**Weaknesses:**

Limited methodological novelty: the framework largely integrates existing techniques (DDIP, LoRA, ADMM) rather than introducing fundamentally new concepts. Evaluation is insufficiently comprehensive—comparisons are limited to FDK, with no benchmarking against recent state-of-the-art reconstruction or diffusion-based methods. Heavy reliance on synthetic phantoms for training raises concerns about generalization to real-world or clinical data. Lack of external validation or reader studies makes it difficult to assess practical or clinical impact. The method is computationally complex, but no analysis of runtime, scalability, or resource requirements is provided.

**Justification Of Rating:**

see above

---

### Decision · Program_Chairs · 2026-05-08

Accept (Poster)